# Elicitation of Hyoscyamine Production in *Datura stramonium* L. Plants Using Tobamoviruses

**DOI:** 10.3390/plants11233319

**Published:** 2022-12-01

**Authors:** Daniel Mihálik, Richard Hančinský, Šarlota Kaňuková, Michaela Mrkvová, Ján Kraic

**Affiliations:** 1National Agricultural and Food Centre, Research Institute of Plant Production, Bratislavská cesta 122, 92168 Piešt’any, Slovakia; 2Faculty of Natural Sciences, University of Ss. Cyril and Methodius, Nám. J. Herdu 2, 91701 Trnava, Slovakia

**Keywords:** thorn apple, tropane alkaloid, in vivo, virus infection, TMV, ToMV, PMMoV

## Abstract

*Datura stramonium* L. produces tropane alkaloids, and the hyoscyamine is dominant among them. Hyoscyamine is produced by hairy root cultures in vitro derived from native plants or plants with the genetically modified biosynthetic pathway for hyoscyamine. A common procedure is extraction from cultivated plants. Elicitors for increased production can be used in both cases. Live viruses are not well known for use as elicitors, therefore, *D. stramonium* plants grown in soil were artificially infected with the tobamoviruses Pepper mild mottle virus (PMMoV), Tomato mosaic virus (ToMV), and Tobacco mosaic virus (TMV). Differences in the content of hyoscyamine were between capsules and roots of infected and non-infected plants. Elicitation increased content of hyoscyamine in capsules 1.23–2.34 times, compared to the control. The most effective viruses were PMMoV and ToMV (isolate PV143), which increased content to above 19 mg/g of fresh weight of a capsule. The effect of each virus elicitor was expressed also in hyoscyamine content in roots. Elicited plants contained 5.41–16.54 times more hyoscyamine in roots compared to non-elicited plants. The most effective elicitor was ToMV SL-1, which raised production above 20 mg/g fresh weight of roots. It has been shown that tobamoviruses can be used as biotic elicitors.

## 1. Introduction

*Datura stramonium* L. is known under different names (thorn apple, jimsonweed, devil’s snare, devil’s trumpet). It is an annual herb from the Solanaceae family, native to the southwest of the USA, Mexico, Central America, and the Greater Antilles. Currently, it is spread worldwide and belongs to the cosmopolitan weed species [1]. The whole plant, including the seeds, is poisonous. The plant can be dangerous, especially because of possible contamination of livestock feed. *D. stramonium* L. contains a variety of alkaloids, saponins, tannins, steroids, flavonoids, phenols, and glycosides [2]. Twenty-five tropane alkaloids were identified in its roots, leaves, and seeds using gas chromatography-mass spectrometry (GC-MS) [3]. Later, the number of identified tropane alkaloids increased to more than sixty [4]. The most interesting among them are hyoscyamine, atropine, and scopolamine. These compounds potently regulate the human central nervous system. The hyoscyamine has remarkable effects in the treatment of Parkinson’s disease, bradyarrhythmias, genitourinary symptoms, and others [5]. Hyoscyamine is the major tropane alkaloid in *D. stramonium* L. [3,6], is a precursor for the synthesis of scopolamine, and an enantiomer of atropine that is formed by racemization during the extractive procedure.

*D. stramonium* is an interesting plant species due to medicinal applications of tropane alkaloids. The demand for tropane alkaloids can be fulfilled by chemical synthesis or by biotechnological production. They can be produced biotechnologically using in vitro cell suspension cultures derived from *D. stramonium* L. [7] or root cultures established from native or transgenic hairy roots [8]. An alternative is the heterologous production of tropane alkaloids in transgenic yeast [9]. Another option is extraction of hyoscyamine directly from plants cultivated in the greenhouse or in the field. *D. stramonium* grows well in a wide range of climate, from tropical to temperate. Production of tropane alkaloids in plants of *D. stramonium* L. starts two weeks after seed germination. Their quantity reaches maximum at the end of the tenth week after seed germination, then gradually decreases as the plants entered the generative phase [10]. Many factors such as plant ploidy, age, light and water conditions, exogenously applied chemicals (fertilisers, hormones) as well as environmental conditions (location, climate, altitude, precipitation, insects) affect content and composition of alkaloids [11]. Irrigation of cultivated plants and soil moisture can simply and effectively increase the production of specific tropane alkaloids [12]. A higher level of accumulation of the desired alkaloids can therefore be achieved by regulating the conditions and optimizing the technological process for plant growing. There is also the possibility to stimulate and increase production of secondary metabolites, including tropane alkaloids. The in vitro cell and tissue cultures derived from relevant species, including the genus *Datura*, can be elicited by various elicitors for enhanced production of tropane alkaloids, including hyoscyamine [13]. It can be effectively produced by in vitro hairy root cultures, especially if elicitors are applied [13,14]. Results that present elicitation of cultivated plants in vivo are very limited in number, although it should be less technically demanding. Elicitor in solution can be applied by foliar spraying of the living plants, watering to the roots, or by swelling into the seeds before planting. Elicitation in vivo should be a gentle tool in influencing the accumulation of secondary metabolites in plants, but the result is dependent on more than only the elicitor compound and its concentrations [15,16,17,18]. The elicitation of increased production of tropane alkaloids in *D. stramonium* plants by artificial infection with a live virus has not been described. Verification of such a hypothesis could reveal the potential of such an approach for increased production of tropane alkaloids. Therefore, the aim of the study was to elicit plants of *D. stramonium* in vivo by artificial infection with tobamoviruses and analyse the content of the dominant tropane alkaloid hyoscyamine in capsules and roots.

## 2. Results

Artificial infections of plants performed in vivo with PMMoV, TMV, and three isolates of ToMV affected differently the biosynthesis of tropane alkaloids and content of analysed hyoscyamine. All three tobamoviruses and their strains, used as biotic elicitors, statistically significantly (*p* < 0.05) increased content of hyoscyamine in comparison with the control, the non-elicited plants. This was confirmed in the content of hyoscyamine analysed in extracts from capsules, as well as from roots of elicited plants (Figure 1).

The values of hyoscyamine in the capsules of the elicited plants, compared to the control, increased 1.23–2.34 times, depending on the tobamovirus used. Concentration of hyoscyamine in capsules of control plants was 8.513 ± 0.877 mg/g DW (grams of fresh weight). The mildest increase (statistically non-significant at *p* < 0.05) in hyoscyamine production to an average value of 10.442 ± 1.653 mg/g DW was induced by ToMV isolate SL-1. Conversely, the most effective elicitors were PMMoV and ToMV (isolate PV143), which increased hyoscyamine production to mean values of 19.727 ± 1.799 mg/g DW and 19.945 ± 0.938 mg/g DW, respectively (Figure 1). This was 2.32–2.34 times more than in the control, non-elicited plants. Figure 2 compares the content of hyoscyamine in analysed control plants (Figure 2A) and in the plants elicited with PMMoV (Figure 2B), where the content of hyoscyamine was the most elevated.

The effect of virus infections was much more evident in the hyoscyamine content in roots of elicited plants (Figure 1). Non-elicited control plants contained 1.223 ± 0.078 mg/g DW of hyoscyamine in root extracts. All tobamoviruses caused an increase in hyoscyamine content after artificial infection. The increase in hyoscyamine content in the roots of the elicited plants was 5.41–16.54 times, compared to roots of control plants. Elicitation statistically significantly (*p* < 0.05) enhanced the level of hyoscyamine in roots to at least a value of 6.619 ± 0.406 mg/g DW (elicited by ToMV P143). The most effective elicitor was ToMV SL-1, which raised production up to 20.232 ± 1.240 mg/g DW. Comparison of the hyoscyamine content in analysed control plants and in the plants elicited with ToMV SL-1, where the content of hyoscyamine was the most elevated, is presented in Figure 3.

All three live tobamoviruses (ToMV, PMMoV, TMV) used as biotic elicitors, effectively enhanced production of hyoscyamine in capsules and roots of treated *D. stramonium* plants.

## 3. Discussion

Different biotic and abiotic elicitors that can regulate the production of the secondary metabolites have been reported in different plant species [19]. The biotic elicitors have biological origins and are derived from the plant pathogen or from the plant itself [20]. The most frequently used pathogens are fungi and bacteria [19]. However, microbial biotic elicitors could also include plant viruses. Interactions with viruses induce in plants positive or negative effects on plant metabolism and alter the synthesis of many plant hormones [21]. Some viruses or their components, such as coat proteins, were identified as specific elicitors of R-genes, playing an important role in the controlling of diseases related to plant viruses [22]. Viruses induce a variety of responses in host plant cells, ranging from non-specific changes in gene expression to specific interactions between virus and host proteins [23].

A significant role in plant defence against the virus infection is played by the secondary metabolites synthesised by the plant. Nevertheless, results published on this topic are very rare. Presence of viral infection often decreased the content of produced secondary metabolites. This has been observed in virus-infected plants of *Echinacea purpurea* L. (Moench.) [24], *Crocus sativus* L. [25], and *Humulus lupulus* L. [26]. The virus also acted in the same way in *D. stramonium* plants after artificial infection with Potato virus X (PVX) [27] and Potato virus Y (PVY) [28]. Content of alkaloids, including atropine, was decreased in infected plants. The decrease in alkaloid content in plants inoculated with PVX was assigned to the deviation in metabolic pathways of alkaloids. The virus usually caused the opposite effect to the elicitor in these cases.

On the contrary, infection by different viruses increased production of a group of substances such as total polyphenols and flavonoids in *Passiflora edulis* [29] and flavonols in *Vitis vinifera* [30]. There are very few cases published on the increased production in industrially interesting specific metabolites by virus infection. An ambiguous result was published for the production of alkaloids in *Papaver somniferum* plants infected with the Poppy mosaic virus [31].

Live plant viruses are generally not used as elicitors of plant secondary metabolite production, neither in plants in vivo nor in cultured cells, tissues, or organs in vitro. This is especially true when the plant is infected with the virus artificially. Tobamoviruses attack plants from the Solanaceae family, including the natural host *D. stramonium*, which is a useful experimental host for detection, identification, maintenance, or manipulation of viruses. D. stramonium can be a reservoir for viruses, causing economic losses in crops [32]. It is even used as an indicator plant for ToMV [33].

The content of hyoscyamine after elicitation increased several times compared to the control. It reached significantly higher values than are generally achieved in hairy root cultures in vitro either elicited or non-elicited. There is production usually in the tenths of milligrams per gram of fresh weight [34,35] or in milligrams per gram of dry weight of the plant material [13,36,37], respectively. The content of hyoscyamine or atropine, respectively, in cultivated plants of D. stramonium varies depending on the developmental stage and used an organ of plant [6,38], and geographical origin [39,40].

All three tobamoviruses used in this study for the elicitation of hyoscyamine production in D. stramonium plants proved to be effective and promising biotic elicitors. However, their use is more complex compared to the application of standard biotic or abiotic elicitors. Generally, viral elicitors have not been sufficiently studied so far. Studying their interactions with plants in terms of their ability to elicit the production of interesting secondary metabolites could bring new insights and practical applications.

## 4. Materials and Methods

### 4.1. Plant Material and Virus Isolates

The sample of *Datura stramonium* L. seeds was obtained from a plant seed supplier company SemenaOnline s.r.o. (Prague, Czech Republic). Before germination, the seeds were scarified for 20 min in concentrated sulphuric acid, followed by washing several times in tap water. After short air drying, seeds were stratified in moist sterile sand for 3 months at 4 °C and sown in sterile soil in plastic containers and were cultivated in a growth chamber under controlled conditions (~55.5 ± 5 μmol.m^−2^.s^−1^ photon flux density, 10 h/14 h light/dark photoperiod, and 22 °C/20 °C day/night temperature). Seeds originating from healthy plants were, after the same treatment, again sown and cultivated under an insect-proof screen-house in the Research Institute of Plant Production (Piešťany, Slovakia). Only properly developed plants with the correct and consistent habitus were used for further experiments.

Five isolates of three tobamoviruses (Table 1) were maintained alive in inoculated experimental plants. Virus presence in plants was tested by immunoblotting analysis using a polyclonal antibody developed for detection of tobamoviruses [41]. The symptomatic leaves from virus-positive plants were collected, sliced, mixed, and deep-frozen as a virus inoculum for subsequent experiments.

### 4.2. Plant Inoculation

Forty-five days after germination, *D. stramonium* L. plants with 3–4 fully expanded leaves were mechanically inoculated with tobamovirus isolates. The inoculum was prepared by grinding of 0.5 g of deep-frozen virus-infected leaves in the Norit buffer, pH 7.0 (0.05 M sodium/potassium phosphate buffer, pH 7.0, 1 mM ethylenediaminetetraacetic acid, 5 mM sodium diethyldithiocarbamate, 5 mM thioglycolic acid) with a ratio of 1:10 (*w*/*v*). Plants were mechanically inoculated with virus inoculum by rubbing on leaves dusted with carborundum inoculum. Ten minutes after inoculation, the inoculated leaves were sprayed with sterile water to remove the inoculum and carborundum. Plants inoculated in the same manner with the Norit buffer were used as controls. Twenty-one days after inoculation, the presence of viral protein was tested in apical leaves using the immunoblotting technique. The roots and capsules from virus-positive and control plants were harvested and deep-frozen at −80 °C.

### 4.3. Hyoscyamine Analysis

Sample extraction was performed with minor modifications according to a previously published protocol [43]. Deep-frozen roots and capsules were ground using the TissueLyser II (Qiagen GmbH, Hilden, Germany) and dried at 45 °C. Fifty milligrams of a sample were extracted with 6 mL of hexane for 5 min. The hexane phase containing fat compounds was evaporated and 12 mL of 0.1 M HCl was added for 10 min. After centrifugation, the solution was adjusted to pH 10.0 with 28% NH_4_OH and three times extracted with an equal volume of CHCl_3_. The combined chloroform extracts were dried over anhydrous Na_2_SO_4_, filtered, and evaporated in a vacuum to give the crude alkaloid fractions. Obtained residues were suspended in 1 mL of CH_2_Cl_2_ and filtrated through a 0.2 μm filter.

Analysis of hyoscyamine in control and elicited *D. stramonium* L. plants was carried out using gas chromatography-mass spectrometry (GC-MS) according to the previously published method [44] using the Agilent 7890B GC equipped with Agilent 5977A Series GC/MSD (Agilent Technologies, Santa Clara, CA, USA). Hyoscyamin was analysed using a low-bleed VF-5 MS column (30 m × 0.25 mm × 0.25 µm) (Agilent Technologies, Santa Clara, CA, USA). One microliter-aliquots of the sample were injected in split-less mode into the GC column by a HP 7683 Automatic liquid sampler (Agilent Technologies, Santa Clara, CA, USA) at a constant flow rate of helium of 1 mL/min used as the carrier gas. Temperature of the injector was operated isothermally at 250 °C, temperature of the oven was kept at 40 °C for 1 min, increased by 30 °C min^−1^ to 130 °C, then by 10 °C min^−1^ to 280 °C, and kept at 280 °C for 5 min. The temperatures of the transfer line and ion source were 300 °C and 200 °C, respectively. Ions were generated by a 70 eV electron beam. Mass spectra were recorded with a scanning range of 30–600 m/z, at 4.7 scans.s^−1^ [44]. The solvent delay time was set at 3 min.

A standard solution of hyoscyamine sulphate (CAS 620-61-1, PhytoLab GmbH & Co. KG, Vestenbergsgreuth, Germany) was used for the identification and quantification of analysed samples. The calibration curve was created by solution of the standard dissolved in CH_2_Cl_2_ at seven concentrations (0.01, 0.02, 0.03, 0.075, 0.4, 0.8, and 1.6 mg/mL). Identification of target alkaloid was also confirmed by comparing the mass spectra of analysed samples to those of commercial spectra in the NIST 2007 library databases.

GC-MS data acquisition was carried out by the MSD ChemStation Data Analysis (Agilent Technologies, Santa Clara, CA, USA). The peak area in a GC-MS chromatogram was automatically integrated and corrected through the Agilent ChemStation software (Agilent Technologies, Santa Clara, CA, USA). The hyoscyamine was identified by comparing the mass spectra of samples to those of a standard solution of hyoscyamine sulfate (CAS 620-61-1, PhytoLab GmbH & Co. KG, Vestenbergsgreuth, Germany) and searching in the NIST 2007 library databases, with the assistance of their qualifier ions. The value m/z of analysed hyoscyamine was 124.1.

### 4.4. Statistical Analysis

Obtained data were evaluated by the analysis of variance (ANOVA) using the Statgraphics software version 19.2.01 (Statgraphics Technologies, Inc., The Plains, VA, USA). Significant differences were compared using the least significant difference (LSD) test at 5% level of significance (*p* < 0.05). The variables with significant differences (*p* < 0.05) between the control and compared groups were marked with a letter from the alphabet. All experiments were carried out in three replications for each treatment.

## 5. Conclusions

Artificially infecting *D. stramonium* L. plants with live tobamoviruses has been shown to be an efficient way to elicit hyoscyamine production. In the analysed extracts from the capsules and roots of the elicited plants, its content increased several times. The increase in the content of hyoscyamine in the roots was statistically significant (at *p* < 0.05) after elicitation by all used viruses and their strains, respectively. Hyoscyamine content in the capsules also increased statistically significantly (at *p* < 0.05) after elicitation with viruses except for one, the ToMV SL01. ToMV SL01 increased production to the level on the threshold of statistical significance. Regarding the interaction of a virus and a natural plant host naturally producing interesting secondary metabolites mediated by artificial infection, the procedure is technically relatively simple. It also allows us to consider the use of viruses, and respectively their components, in production systems in vitro.

## Figures and Tables

**Figure 1 plants-11-03319-f001:**
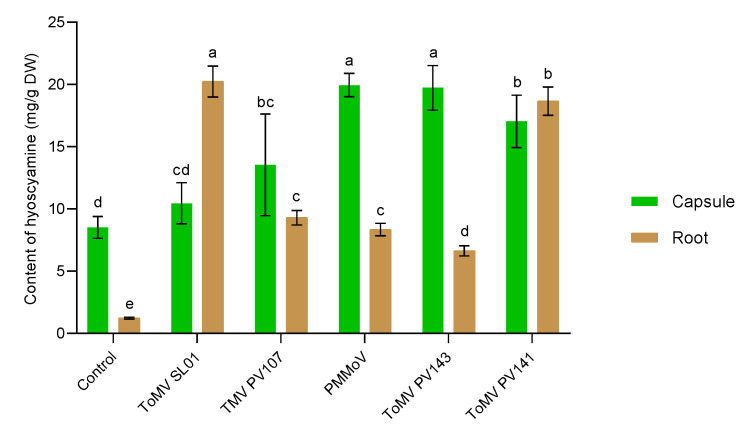
Content of hyoscyamine in capsules and roots of control and elicited plants of *D. stramonium*. Different superscript letters indicate a significant difference at *p* < 0.05.

**Figure 2 plants-11-03319-f002:**
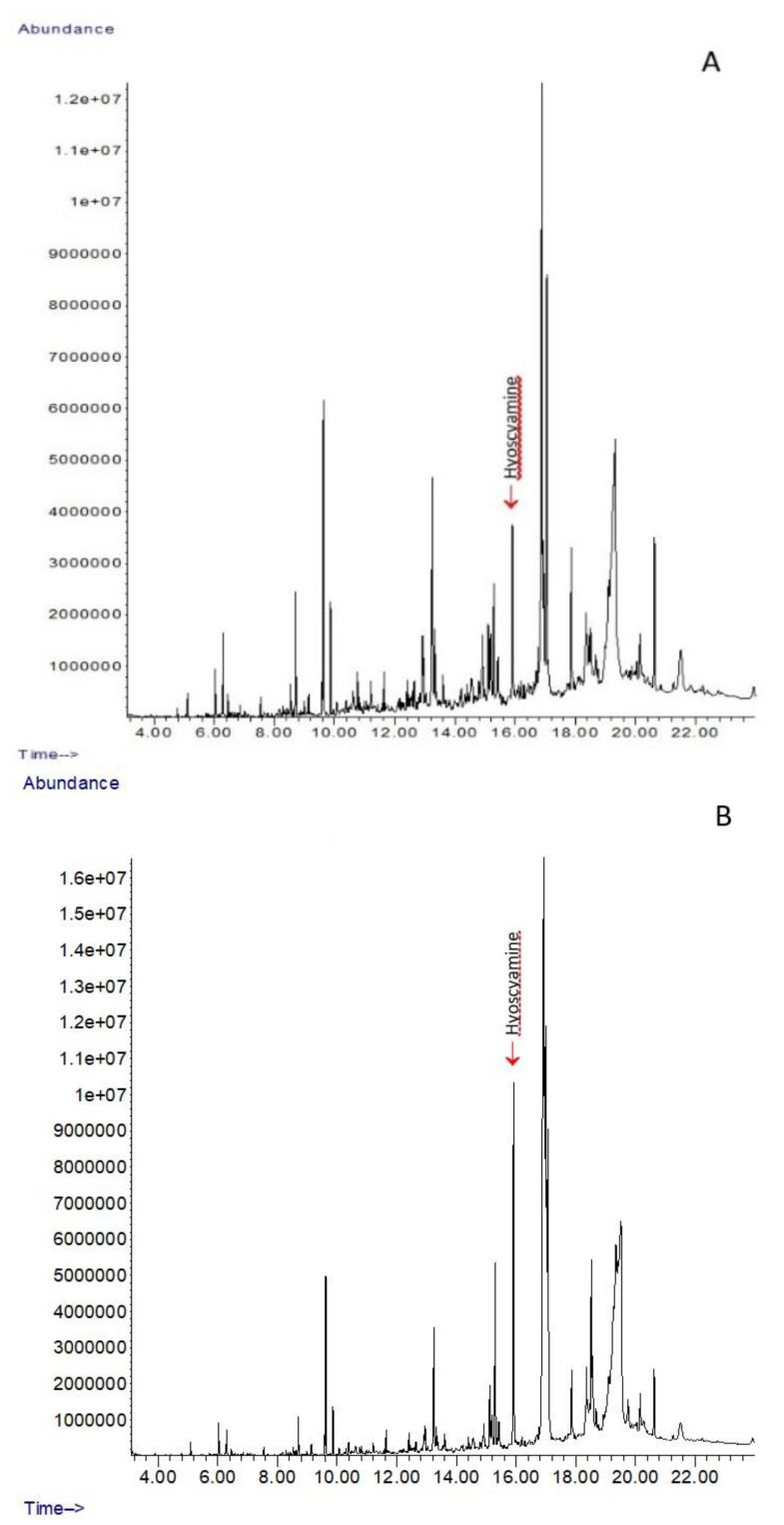
Chromatograms (detector response—abundance vs. retention times in min) obtained from the GC-MS analysis of *D. stramonium* extracts from capsules. (**A**) Healthy control plants, (**B**) Plants elicited with PMMoV. The peak representing hyoscyamine is indicated.

**Figure 3 plants-11-03319-f003:**
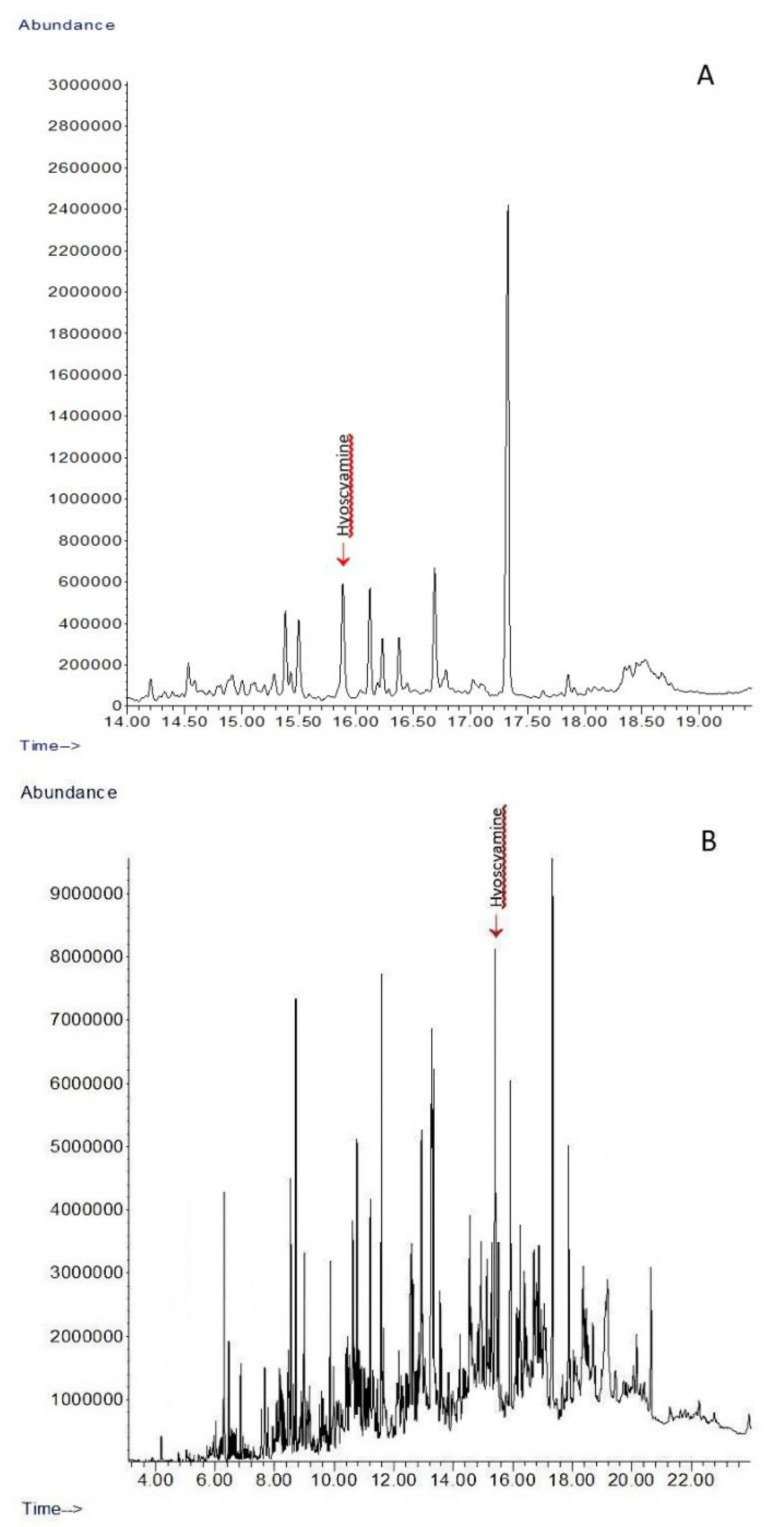
Chromatograms (detector response—abundance vs. retention times in min) obtained from the GC-MS analysis of *D. stramonium* extracts from roots. (**A**) Healthy control plants, (**B**) Plants elicited with ToMV SL-1. The peak representing hyoscyamine is indicated.

**Table 1 plants-11-03319-t001:** Tobamovirus isolates used for inoculation of *D. stramonium* L. plants.

Virus	Isolate	Reference	Experimental Hosts
PMMoV	SK2	GenBank: ON493797 (Submitted, waiting for publishing)	*Capsicum annum* L.
TMV	PV-0107	DSMZ no. PV-0107	*Solanum lycopersicum* L.
ToMV	SL-1	[42], GenBank: KY912162.1	*Solanum lycopersicum* L. cv. Monalbo
	PV-0141	DSMZ no. PV-0141	*Solanum lycopersicum* L. cv. Monalbo
	PV-0143	DSMZ no. PV-0143	*Solanum lycopersicum* L. cv. Monalbo

## Data Availability

Not applicable.

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
