# Peer review of "Elicitation of Hyoscyamine Production in Datura stramonium L. Plants Using Tobamoviruses"

_plants, 2022, doi:10.3390/plants11233319_

Round 1
Reviewer 1 Report
I have one concerning comment regarding how we can use plant viruses as an elicitor. I think this is can not be applicable, especially under open field conditions.
Reviewer 2 Report
In this manuscript Live viruses are used as elicitors in D. stramonium plants grown in soil which were artificially infected with the tobamoviruses Pepper mild mottle virus (PMMoV), Tomato mosaic virus (ToMV), and Tobacco mosaic virus (TMV). Differences in the content of hyoscyamine were between capsules and roots of infected and non-infected plants. Elicitation increased content of
hyoscyamine in capsules 1.23–2.34 times, compared to the control. The most effective viruses were reported PMMoV and ToMV (isolate PV143), which increased content to above 19 mg/g of fresh weight of ther capsule. good charts and results are consistent with the research done
Comments check English language
It would be good to add recovery of hiosciamine in the biological material as well as LOD LOQ and other common paprameters for validation of measurements by GC and internal standard
Reviewer 3 Report
I have read with much interest the MS submitted. The design is appropriate and interpretation is supported with data. The research gap has been properly identified. I have some concerns that needs to be addressed:
In abstract: What is the difference between: ''Tomato mosaic virus (ToMV), and Tobacco mosaic virus (TMV).' !
I would suggest more data/figures in the abstract.
e.g. Differences in the content of hyoscyamine were between capsules and roots of infected and non-infected plants should be support with stats + data
Please refer to lines 19-20 in abstract ‘’ increased content of hyoscyamine in capsules 1.23–2.34 times. Was it significant?
In section 4.1. Please elaborate how many plants were included in the sample.
Elaborate how authors identified plants were ‘’healthy’’- L179
